# Public Opinion Field Effect Fusion in Representation Learning for Trending Topics Diffusion

**Junliang Li**[1,2], **Yajun Yang**[1,2]*, **Qinghua Hu**[1], **Xin Wang**[1], **Hong Gao**[3]

[1]College of Intelligence and Computing, Tianjin University, China
[2]State Key Laboratory of Communication Content Cognition, Beijing, China
[3]College of Mathematics and Computer Science, Zhejiang Normal University, China
[1]{lijunliang,yjyang,huqinghua,wangx}@tju.edu.cn,[3]honggao@zjnu.edu.cn

## Abstract

Trending topic diffusion and prediction analysis is an important problem and has been well studied in social networks. Representation learning is an effective way to extract node embeddings, which can help for topic propagation analysis by completing downstream tasks such as link prediction and node classification. In real world, there are often several trending topics or opinion leaders in public opinion space at the same time and they can be regarded as different centers of public opinion. A public opinion field will be formed surrounding every center. These public opinion fields compete for public's attention and it will potentially affect the development of public opinion. However, the existing methods do not consider public opinion field effect for trending topics diffusion. In this paper, we introduce three well-known observations about public opinion field effect in media and communication studies, and propose a novel and effective heterogeneous representation learning framework to incorporate public opinion field effect and social circle influence effect. To the best of our knowledge, our work is the first to consider these effects in representation learning for trending topic diffusion. Extensive experiments on real-world datasets validate the superiority of our model.

## 1 Introduction

With the rapid development of the Internet, hundreds of millions of people access and share information on social medias. Social networks have become the most important platform for producing and spreading public opinions. Various trending topics widely spread on social networks will bring a series of consequences. For example, fake news quickly spreads on social networks, will cause governments or organizations to fall into a huge crisis of public opinion [11]. Studying the diffusion and prediction of trending topics can effectively help governments and enterprises to timely discover potential public opinion crisis and deal with it to avoid their reputation and economic losses [52].

The existing studies on trending topic diffusion are mainly divided into two categories: content-based and network-based. The content-based methods [54, 21, 55, 26] aim to predict the dissemination trends, e.g., number of comments and retweets, mainly for a blog based on content features such as tags of text, images, and videos, but not cosider the network features. For network-based methods, various prediction model based on graph embedding techiniques [6, 53, 34, 62] or random walk [33, 51, 2, 22] are proposed to calculate the information propagation probability for every node in social networks. Recently, representation learning on heterogeneous graph becomes an effective way to extract a low-dimensional node embedding by considering the network structure and node attribute features, which can be applied to downstream tasks of propagation analysis such as link prediction and node classification [18, 28, 27, 61, 30].

---

*Corresponding author

37th Conference on Neural Information Processing Systems (NeurIPS 2023).

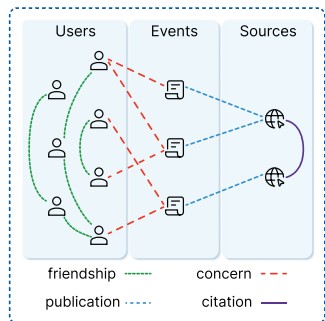 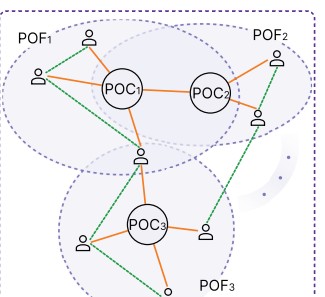 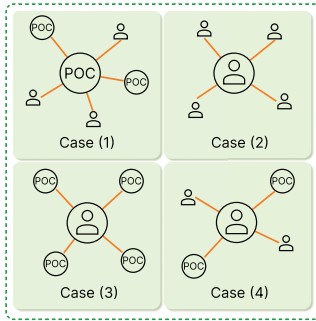

(a) Public opinion network.    (b) An example of several pocs and their pofs.    (c) Four cases for nodes in PON.

Figure 1: Public opinion field effect.

However, the existing representation learning models only consider the information of social network itself but neglect the influence of "public opinion field effect" for trending topic diffusion. In real life, there are often several trending topics or opinion leaders in the public opinion space at the same time. Every topic or opinion leader with the people actively following it in a social network will form a public opinion field. Research on media and communication has shown that the total amount of public's attention is always limited [29]. These public opinion field compete for public's attention and such competition will potentially affect the development of public opinion. Obviously, a topic with higher popularity may attract more attention from people who are following other topics. The existence of public opinion field effect makes the existing representation learning model cannot work well on the trending topic diffusion. A few works on media and communication [6, 53, 34] only present some statistics of public opinion field, e.g., the number of people who talk about different topics, but not consider to incorporate the public opinion field effect into representation learning for trending topic diffusion. Moreover, in real world, people are always involved in a topic propagation if their social circle is actively talking about the same topic, even though they are not interested in this topic at the beginning. This effect is also not considered in existing representation learning model.

In this paper, we study how to incorporate public opinion field effect in representation learning. To the best of our knowledge, our work is the first one regarding this problem for trending topic diffusion. Our proposed model includes "topic propagation representation" and "social circle influence representation". In topic propagation representation, three kinds of nodes, event nodes, information source nodes and opinion leader nodes, are regarded as the centers of public opinion for their corresponding topics. We define public opinion field based on center node and propose a kind of paradigm to quantify the "energy" of public opinion field, which can reflect the attraction power of public opinion field. We introduce three well-known observations about public opinion effect in media and communication studies, "***Attention Competition Effect***", "***Popularity Dominance Effect***" and "***Negative Bias Effect***", and design the model to obtain the higher quality representation by considering these effects in trending topic diffusion. In social circle influence representation, the social circle influence effect is considered in representation learning. The main contributions are summarized below:

(1) We formalize public opinion field effect and propose a novel and effective heterogeneous representation learning model by incorporating public opinion field effect for the first time.

(2) We consider the social circle influence, which extensively exists in real world, to enhance the quality of node representation.

(3) We conduct extensive experiments on three real-world datasets. The experimental results demonstrate the superiority of our method.

## 2   Formalization of Public Opinion Field Effect

In order to better investigate the diffusion of public opinion events in social network, we introduce the "***public opinion field effect***" in representation learning for the first time. In real applications, the public

opinion space includes various public opinion events attracting users' attention in social network. A public opinion field can be regarded as a public opinion event with the users paying attention to it or a high influence user with his/her followers. The interaction of different public opinion fields influences the development of public opinion events. To formalize the public opinion field, we first introduce **Public Opinion Network** (PON), which is essentially a heterogeneous network consisting of different kinds of entities and links. All of the entities in a PON can be categorized into three types: (i) users in the social network, (ii) public events and (iii) sources that publish the events. All of the links in a PON can be categorized into four types: (i) friendship link (user to user), (ii) concern link (user to event), (iii) publication link (source to event) and (iv) citation link (source to source). The definition of PON is given below:

**Definition 1** (Public Opinion Network): A public opinion network (PON) is a heterogeneous directed graph $G = (V, E)$, where $V = U \cup I \cup S$ is the set of nodes and $E \subseteq (U \times U) \cup (U \times I) \cup (S \times I) \cup (S \times S)$ is the set of links among the nodes in $V$. Note that $U$, $I$ and $S$ are disjoint sets of the nodes representing the users in social network, public opinion events and sources publishing the events respectively. □

Fig. 1(a) shows an example of a public opinion network. A public opinion network depicts the public opinion space in a social network. In a public opinion network, every event node, source node and opinion leader node is also called a "public opinion center node", or poc node for simplicity. Note that opinion leader nodes identifying in social networks has been well studied and thus we do not discuss it, but instead consider them as input to our model in this paper. We use $U_h$ and $C$ to denote the sets of highly influential user nodes and poc nodes respectively, i.e., $C = I \cup S \cup U_h$. Next we give the definition of **Public Opinion Field**.

**Definition 2** (Public Opinion Field): Given a public opinion network $G$ and a public opinion center node $v_i \in C$, a public opinion field (or pof for simplicity) $P_i$ of $v_i$ is defined as an induced subgraph of $G$ on $N_i \cup \{v_i\}$, where $N_i$ is the neighbor set of $v_i$ in $G$. □

Fig. 1(b) illustrates an example of several poc nodes and their corresponding pofs in a PON. Note that in a social network a user may be in several distinct pofs at the same time. A poc node can be regarded as the center of "whirlpool" of a public topic and a pof can be regarded as a view in $G$ focusing on a poc node with the users pay attention to it. Existing works neglect "public opinion field effect" in representation learning for public event diffusion. However, in practice, the interaction between different pofs always affects the development of public opinion. There are three well-known observations in media and communication studies as follows:

**Observation 1** (*Attention Competition Effect*): The popularity of news decreases with the number of competing items that are simultaneously available. A user's breadth of attention remains essentially constant and the diversity of memes to which a user can pay attention is bound [29].

**Observation 2** (*Popularity Dominance Effect*): In public opinion space, the topics with the higher popularity are more powerful to attract users' attention [45].

**Observation 3** (*Negative Bias Effect*): The negative events, e.g., fake news or rumors, are easier to spread in social networks [20].

It is noteworthy that we recognize the existence of other effects (e.g., "Empathy Effect" [4, 50] and "Primacy Effect" [16, 56]) in the real world. However these effects are not closely relevant to the influence of coexisting trending topics on information diffusion. In addition, there also are some other effects (e.g., "Snowball Effect" [24, 25] and "Herd Effect" [23, 14]) in media and communication studies, but the meaning behind these effects are very similar to three effects introduced in this paper and thus we do not discuss them.

Observation 1 indicates the amount of attention in public opinion space is always constant, the attention on a current public topic may be seized by another rising topic, e.g., "Feng Wang Effect"[1] in China. Observations 2 and 3 indicate that trending topics and negative events probably grab more attention in public opinion space. For fusion of above three effects in representation learning, we define "energy" of public opinion field to quantify the attraction power of a poc node in PON $G$.

**Definition 3** (Public Opinion Field Energy): Given a poc node $v_i$ with its pof $P_i$, the energy of $P_i$, denoted as $E(P_i)$, is defined as $E(P_i) = \sum_{v_j \in N_i} w_{i,j} \tau_{i,j}$, where $N_i$ is the neighbor set of poc node

---

[1]Feng Wang is a well-known singer in China. Whenever he announces something important, there are always other more trending topics emerging and competing with Feng Wang's event for users' attention.

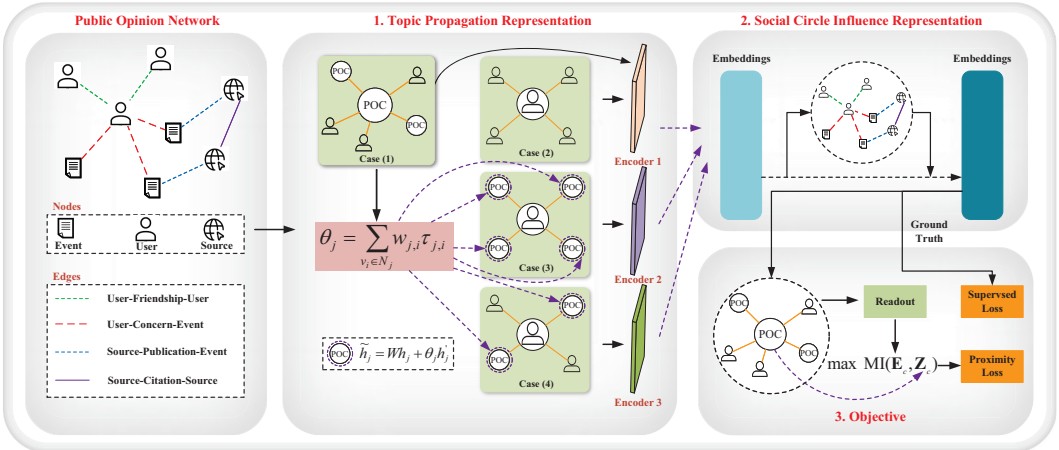

Figure 2: Overall framework of our model.

$v_i$, $w_{i,j} \in \mathbb{R}$ is the popularity contribution of $v_j$ for a poc node $v_i$ and $\tau_{i,j} \in \mathbb{R}$ is the attention degree of $v_j$ for a topic represented by poc $v_i$. □

Here $w_{i,j}$ reflects how much popularity or attention can be brought if $v_j$ participates in the discussion or propagation of public topic $v_i$. It is obvious that Michael Jordan brings more attention than the ordinary person when discussing basketball topics. $\tau_{i,j}$ reflects how much attention that $v_j$ intends to pay on the topic represented by poc node $v_i$.

Note that researchers can calculate $w_{i,j}$ and $\tau_{i,j}$ in any reasonable ways and we utilize attention mechanism function and covariance function to compute them respectively in this paper. We will discuss it in the next section and then prove that attention defined in our representation learning follows Observation 1, i.e., the total amount of attention in the public opinion fields is constant.

If a poc node $v_i$ is related to a negative topic, an energy amplification factor $\eta_i$ $(\eta_i > 1)$ is utilized to amplify the energy of pof $P_i$ by $\eta_i \cdot E(P_i)$ and thus Observation 3 is satisfied in our representation learning framework.

Notably, the reason we use only 1-hop neighbors to construct pof and pof energy is that they are closest to poc. There are also someone know a trending topic indirectly or post their comments on a retweeted content. In fact, this phenomenon has been taken account into our social circle influence representation in Section 3.2. Moreover, our model can be easily extended to handle $k$-hop neighbors by the following equation $E(P_i^{(k)}) = \sum_{t=1}^{k} \sum_{d_{i,j}=t} w_{i,j}\tau_{i,j}$, where $d_{i,j}$ denotes the distance between nodes $v_i$ and $v_j$. We will experimentally illustrate the soundness of the choice of 1-hop neighbors in Section 4.5.

## 3 Representation Learning

We investigate "***topic propagation representation***" and "***social circle influence representation***" in our model. In topic propagation representation, public opinion field effect is introduced to reflect that how users actively participate in trending topics diffusion. However, in practice, some users are passively involved in a trending topic diffusion because most of his/her friends have participated in the discussion about this topic. For example, someone not interested in computer science does not care about the topic about ChatGPT when he/she knows it from news for the first time, but his/her interest will rise if most of his/her friends are talking about ChatGPT and thus he/she will participate in this topic diffusion. We consider such effect in social circle influence representation.

As shown in Fig. 2, our model is divided into three parts: topic propagation representation, social circle influence representation and objective optimization. In topic propagation representation, we first calculate the pof energy, i.e., $\theta_j = E(P_j) = \sum_{v_i \in N_j} w_{j,i}\tau_{j,i}$. Then, pof energy will be combined with the heterogeneous graph convolution module to perform feature aggregation for the four types of nodes in the PON. In social circle influence representation, we combine the representation obtained

by topic propagation and the PON network to enhance the quality of representation learning. Finally, we optimize our model by supervised entropy loss and unsupervised proximity loss.

### 3.1 Topic Propagation Representation

In this section, we introduce how to consider public opinion field effect in topic propagation representation. Because the nodes with different types always have different feature spaces, we first utilize $\boldsymbol{h}_i = \boldsymbol{A}\boldsymbol{x}_i$ to project the features of all the nodes in $G$ into the same space, where $\boldsymbol{x}_i$ is original feature of node $v_i$ and $\boldsymbol{A}$ is a projection matrix. Note that $\boldsymbol{A}$ is different depending on the type of node, i.e., $U$, $I$ and $S$. Next the $w_{i,j}$ and $\tau_{i,j}$ mentioned in Definition 3 can be calculated as follows:

$$w_{i,j} = \lambda_j \cdot \text{att}(\boldsymbol{h}_i, \boldsymbol{h}_j) = \lambda_j \cdot \sigma(\boldsymbol{\Phi}^\top \cdot (\boldsymbol{W}\boldsymbol{h}_i \oplus \boldsymbol{W}\boldsymbol{h}_j)), \ \tau_{i,j} = \sigma(\text{cov}(\boldsymbol{W}\boldsymbol{h}_i, \boldsymbol{W}\boldsymbol{h}_j)) \quad (1)$$

where $\text{att}(\boldsymbol{h}_i, \boldsymbol{h}_j)$ is graph attention mechanism function, which reflects the importance of $v_j$ with respect to poc node $v_i$, $\lambda_j$ is used to measure the influence of $v_j$ in a social network. Previous studies propose various influence measures such as degree centrality, betweenness centrality and so on, and we use normalized degree centrality in this paper. $\boldsymbol{\Phi}$ is the weight vector for performing graph attention, $\boldsymbol{W}$ is a shared linear transformation weight matrix and $\oplus$ is concatenate operation. We select covariance function $\text{cov}(\boldsymbol{x}, \boldsymbol{y}) = (\sum_{i=1}^{f}(x_i - \bar{\boldsymbol{x}})(y_i - \bar{\boldsymbol{y}}))/(f - 1)$, which indicates the correlation between two vectors $\boldsymbol{x}, \boldsymbol{y} \in \mathbb{R}^f$, to measure the amount of attention that $v_j$ intends to pay on the topic represented by poc node $v_i$. It is reasonable since most people prefer to talk about the topics that are more relevant to them.

From Observation 1, we know that the attention amount of every user is limited. If someone intends to pay more attention to a new trending topic, his/her interests in other topics will be dropped off at the same time. As per the discussion, $\tau_{i,j}$ essentially reflects the willingness of $v_j$ to participate in a topic represented by a poc node $v_i$, thus we use a normalization operation on $\tau_{i,j}$ for every user $v_j \in U$:

$$\tau_{i,j} \leftarrow \frac{w_{i,j}\tau_{i,j}}{\sum_{v_i \in C \cap N_j} w_{i,j}\tau_{i,j}} \quad (2)$$

Obviously, normalization operation guarantees that the "***Attention Competition Effect*** (Observation 1)" is satisfied in our representation learning model. Theorem 1 shows that the total amount of attention in public opinion space that is defined in our representation model is constant.

**Theorem 1:** Given a PON $G$, $C$ and $C'$ are any two distinct poc node sets at different times $t$ and $t'$ on $G$ respectively, the amount of attention for the same user set $U$ in $G$ is constant for different times, i.e., $\sum_{v_i \in C, v_j \in U} \tau_{i,j} = \sum_{v_i \in C', v_j \in U} \tau_{i,j}$.

PROOF: Because $\sum_{v_i \in C} \tau_{i,j} = \sum_{v_i \in C'} \tau_{i,j} = 1$, then $\sum_{v_i \in C, v_j \in U} \tau_{i,j} = \sum_{v_i \in C', v_j \in U} \tau_{i,j}$. □

In our representation learning model, a normalization operation is also necessary on $E(P_i)$ to satisfy Observation 3, that is

$$E(P_i) \leftarrow \frac{\eta_i \cdot E(P_i)}{\sum_{v_i \in C} \eta_i \cdot E(P_i)} \quad (3)$$

where $\eta_i > 1$ if poc node $v_i$ is related to negative topic and otherwise $\eta_i = 1$.

All the nodes in PON $G$ can be categorized into four cases: (1) poc nodes; (2) user nodes $v_i$ whose all neighbors are normal user nodes, i.e., $U_1 = \{v_i | v_i \in U \wedge N_i \subseteq U - U_h\}$; (3) user nodes $v_i$ whose all neighbors are poc nodes, i.e., $U_2 = \{v_i | v_i \in U \wedge N_i \subseteq C\}$; and (4) users nodes $v_i$ whose neighbors include both poc nodes and normal user nodes, i.e., $U_3 = \{v_i | v_i \in U \wedge N_i \nsubseteq C \wedge N_i \nsubseteq U - U_h\}$. We give an illustration of all cases in Fig. 1(c) and investigate node representation in trending topics propagation for these four cases respectively.

*Case* (1) and (2): We use the self-attention mechanism to update the features of every poc node $v_i \in C$ and every node $v_i \in U_1$. Specifically, the node representation $\boldsymbol{h}_i'$ of $v_i$ is given below:

$$\boldsymbol{h}_i' = \sigma(\sum_{v_j \in N_i \cup \{v_i\}} \alpha_{i,j} \cdot \boldsymbol{W}\boldsymbol{h}_j), \ \alpha_{i,j} = \text{softmax}(\text{att}(\boldsymbol{h}_i, \boldsymbol{h}_j)) \quad (4)$$

where $\text{att}(\cdot)$ is similar to Eq. (1), $\alpha_{i,j}$ is the attention coefficient between $v_i$ and its neighbor $v_j$. By Eq. (4), we know $\boldsymbol{h}_i'$ of a poc node $v_i$ also can be regarded as a representation of pof $P_i$ because it aggregates all the features of nodes in $P_i$.

*Case* (3): For each user node $v_i \in U_2$, its different poc neighbor nodes will compete for user $v_i$'s attention (***Attention Competition Effect***). The poc neigbor node with the larger pof energy will gain more attention from $v_i$ (***Popularity Dominance Effect***). We consider both self attention mechanism and public opinion field effect in node representation $\boldsymbol{h}_i'$ for $v_i \in U_2$:

$$\boldsymbol{h}_i' = \sigma(\sum\nolimits_{v_j \in N_i \cup \{v_i\}} \alpha_{i,j} \cdot \tilde{\boldsymbol{h}}_j), \ \tilde{\boldsymbol{h}}_j = \boldsymbol{W}\boldsymbol{h}_j + \theta_j \cdot \boldsymbol{h}_j' \tag{5}$$

$$\alpha_{i,j} = \text{softmax}(\text{att}(\boldsymbol{h}_i, \boldsymbol{h}_j, \boldsymbol{\Theta})), \ \text{att}(\boldsymbol{h}_i, \boldsymbol{h}_j, \boldsymbol{\Theta}) = \theta_j \cdot \text{att}(\boldsymbol{h}_i, \boldsymbol{h}_j) \tag{6}$$

Note that $\boldsymbol{h}_j'$ is node representation of poc node $v_j$ calculated in case (1), $\theta_j$ is the normalized pof energy of $v_j$, i.e., $\theta_j = E(P_j)$, and thus $\boldsymbol{\Theta} = (\theta_1, \cdots, \theta_{|C|})$ is the vector of all the normalized pof energy. Specifically, we need to second normalize the pof energy of all poc neighbors of $v_i \in U_2$: $\theta_j \leftarrow \theta_j / \sum_{v_j \in N_i} \theta_j$. In Eq. (5), the term $\theta_j \cdot \boldsymbol{h}_j'$ is utilized to reflect the influence of pof $P_j$ for representation of $v_i$, because $\boldsymbol{h}_j'$ is the representation of $P_i$ as explained in case (1).

*Case* (4): For every node $v_i \in U_3$, it has two kinds of neighbors, poc neighbors $v_j \in C$ and normal user neighbors $v_j \in U - U_h$. To consider heterogeneity, RGCN [44] designs different feature projection matrices for different types of links. Inspired by this, we use two different weight matrices $\boldsymbol{\Phi}_1$ and $\boldsymbol{\Phi}_2$ to calculate two different attention coefficient $\alpha_{i,j}^1$ and $\alpha_{i,j}^2$, that is

$$\alpha_{i,j}^k = \text{softmax}(\text{att}(\boldsymbol{\Phi}_k, \boldsymbol{h}_i, \boldsymbol{h}_j)), \ \text{att}(\boldsymbol{\Phi}_k, \boldsymbol{h}_i, \boldsymbol{h}_j) = \sigma(\boldsymbol{\Phi}_k^\top \cdot (\boldsymbol{W}\boldsymbol{h}_i \oplus \boldsymbol{W}\boldsymbol{h}_j)) \tag{7}$$

where $k = 1$ if $v_j \in S_1 = N_i \cap C$ and $k = 2$ if $v_j \in S_2 = N_i \cap (U - U_h)$. Therefore, the node representation $\boldsymbol{h}_i'$ of $v_i \in U_3$ is given below:

$$\boldsymbol{h}_i' = \sigma(\sum\nolimits_{v_j \in S_1} \alpha_{i,j}^1 \cdot \tilde{\boldsymbol{h}}_j + \sum\nolimits_{v_j \in S_2} \alpha_{i,j}^2 \cdot \boldsymbol{W}\boldsymbol{h}_j), \ \tilde{\boldsymbol{h}}_j = \boldsymbol{W}\boldsymbol{h}_j + \theta_j \cdot \boldsymbol{h}_j' \tag{8}$$

Like case (3), $\boldsymbol{h}_j'$ is node representation for poc node $v_j \in S_1$ calculated in case (1).

## 3.2 Social Circle Influence Representation

As per the discussion, users are always involved in the diffusion of trending topics that their social circle are talking about. To investigate such influence, we propose social circle influence representation after topic propagation process. In social circle influence representation, a user node will be imposed influence from their neighbors who have paid attention to the some trending topic. In PON $G$, every user node in $U$ will receive different amounts of topic information from its user neighbors. We use the graph attention mechanism, $\rho_{i,j} = \text{softmax}(\text{att}(\boldsymbol{h}_i^{'}, \boldsymbol{h}_j^{'}))$, to measure the influence weight for node $v_i$ from its neighbor $v_j$, where $\boldsymbol{h}_i^{'}$ and $\boldsymbol{h}_j^{'}$ are the topic propagation representation of node $v_i$ and $v_j$ in Section 3.1. Therefore, the total influence on user $v_i \in U$ from its social circle (i.e., $N_i$) can be computed as follows:

$$\text{Pool}(\boldsymbol{h}_j^{'}|v_j \in N_i) = \sum\nolimits_{v_j \in N_i} \rho_{i,j} \cdot \boldsymbol{h}_j^{'} \tag{9}$$

Note that even though $v_i$ can be influenced by its neighbors, its own representations learned from topic propagation representation are fundamental and important. Thus the final representation $\boldsymbol{z}_i$ of a user node $v_i \in U$ is as follows:

$$\boldsymbol{z}_i = \sigma(\boldsymbol{W} \cdot (\boldsymbol{h}_i^{'} \oplus \text{Pool}(\boldsymbol{h}_j^{'}|v_j \in N_i))) \tag{10}$$

## 3.3 Optimization

Note that after topic propagation representation and social circle influence representation, the features of all the nodes in $G$ are updated to $\boldsymbol{Z} \in \mathbb{R}^{|V| \times f}$.

***Unsupervised Proximity Loss***. In real world, proximity exists extensively in a public opinion field. The users within the same pof always have similar features such as background, career, interest and so on. For example, users who actively discuss ChatGPT usually have the same background or interest in computer science. Moreover, the features of the nodes within a pof should be correlated with feature of poc node in this pof, which is the reason that they prefer to pay attention to this poc node. In this paper, we utilize such proximity property, by enhancing the correlation of embeddings between poc node and all the nodes within the same pof, to obtain higher-quality embeddings from

our representation learning model. Specifically, we first use the readout function to aggregate the features of all the nodes within pof $P_i$ to generate their common embedding $\boldsymbol{e}_i \in \mathbb{R}^f$:

$$\boldsymbol{e}_i = \text{Readout}(\boldsymbol{H}_i) \tag{11}$$

where $\boldsymbol{H}_i \in \mathbb{R}^{|N_i \cup \{v_i\}| \times f}$ denotes the embedding of all the nodes in the pof $P_i$. Readout$(\cdot)$ denotes a simple permutation-invariant function (e.g., sum and mean). Let $\boldsymbol{E}_c \in \mathbb{R}^{|C| \times f}$ and $\boldsymbol{Z}_c \in \mathbb{R}^{|C| \times f}$ denote the readout embedding of all pofs and the embedding of all poc nodes, respectively. In order to maximize the correlation between the poc node and all the nodes within the same pof, the mutual information (MI) between $\boldsymbol{E}_c$ and $\boldsymbol{Z}_c$ is maximized. Specifically, given two random variables $\boldsymbol{X}$ and $\boldsymbol{Y}$, we adopt MINE method proposed in [3] to maximize the lower bound of $\text{MI}(\boldsymbol{X}, \boldsymbol{Y})$ as follows:

$$\max \text{MI}(\boldsymbol{X}, \boldsymbol{Y}) \geq \mathbb{E}_{P(\boldsymbol{X}, \boldsymbol{Y})}[D(\boldsymbol{X}, \boldsymbol{Y})] - \log \mathbb{E}_{P(\boldsymbol{X})P(\boldsymbol{Y})}\left[e^{D(\boldsymbol{X}, \boldsymbol{Y})}\right] \tag{12}$$

where $D(\boldsymbol{X}, \boldsymbol{Y})$ is a binary discriminator. Therefore, to maximize $\text{MI}(\boldsymbol{E}_c, \boldsymbol{Z}_c)$, the unsupervised proximity loss $L_p$ is defined below:

$$L_p(\boldsymbol{E}_c, \boldsymbol{Z}_c) = -\mathbb{E}_{P(\boldsymbol{e}_i, \boldsymbol{z}_i)}[D(\boldsymbol{e}_i, \boldsymbol{z}_i)] + \log \mathbb{E}_{P(\boldsymbol{e})P(\boldsymbol{z})}\left[e^{D(\boldsymbol{e}_i, \boldsymbol{z}_i)}\right] \tag{13}$$

where $\boldsymbol{e}_i$ and $\boldsymbol{z}_i$ denote the readout representation and the original representation of the poc node $v_i$. The discriminator $D(\cdot, \cdot)$ is defined as a simple neural network, $D(\boldsymbol{e}_i, \boldsymbol{z}_i) = \sigma(\boldsymbol{e}_i^\top \boldsymbol{W} \boldsymbol{z}_i)$. In the experiment, for each batch, we estimate the first term in Eq. (13) by selecting a set of $\{(\boldsymbol{e}_i, \boldsymbol{z}_i)\}_{i=1}^B$ from the joint distribution $P(\boldsymbol{e}_i, \boldsymbol{z}_i)$, and then we shuffle the $\boldsymbol{z}_i$ in the batch and generate "negative pairs" in order to estimate the second term.

***Supervised Loss.*** For downstream tasks, we can minimize the cross-entropy loss function between ground truth and prediction:

$$L_s = -\sum_{l \in L} \boldsymbol{Y}^l \ln(\hat{\boldsymbol{C}} \cdot \boldsymbol{Z}^l) \tag{14}$$

where $\hat{\boldsymbol{C}}$ is the parameter of the classifier, $L$ is the set of labeled nodes/edges, $\boldsymbol{Y}^l$ and $\boldsymbol{Z}^l$ are the label vector and embeddings of the labeled nodes/edges respectively. Finally, We define the total loss by linearly combining these two component losses: $L = L_s + \mu L_p + \varphi L_{reg}$, where $\mu$ is the hyperparameter that controls the proportion of unsupervised proximity loss and $\varphi L_{reg}$ is the regularization term to alleviate overfitting.

## 4 Experiments

### 4.1 Experimental Setup

***Datasets.*** Detailed statistical information of the datasets is shown in Table 1. (1) **BuzzFeed and PolitiFact**. The two datasets are from a comprehensive fake news detection benchmark dataset, FakeNewsNet [47, 48]. They are collected from two platforms with fact-checking: BuzzFeed and PolitiFact, both containing news content with labels and social context information. (2) **Twitter**. This dataset [31, 47] contains multiple articles, with each article corresponding to one event. For each event, the dataset includes its source, a list of participating users, and their tweets. This dataset also includes Twitter profile description and the list of Twitter profiles each user follows. More information on the three datasets can be found in ***Appendix*** A.1 and ***Appendix*** A.2.

Table 1: Statistics of datasets.

| Dataset | Relation (A-B) | # A | # B | # A-B | Average Degree | Average Clustering Coefficient | Connectivity |
|---|---|---|---|---|---|---|---|
| BuzzFeed | user-user | 15257 | 15257 | 634750 | 88.01 | 0.087 | True |
| | user-event | 15257 | 182 | 22779 | | | |
| | source-event | 28 | 182 | 174 | | | |
| PolitiFact | user-user | 23865 | 23865 | 574744 | 53.13 | 0.066 | True |
| | user-event | 23865 | 240 | 32791 | | | |
| | source-event | 17 | 240 | 236 | | | |
| Twitter | user-user | 54461 | 54461 | 3172794 | 118.25 | 0.094 | True |
| | user-event | 54461 | 1054 | 75809 | | | |
| | source-event | 442 | 1054 | 1054 | | | |
| | source-source | 442 | 442 | 3707 | | | |

***Baselines.*** For simplicity, our model is denoted as POFD (**P**ublic **O**pinion **F**ield Effect Fusion in Representation Learning for Trending Topics **D**iffusion). We compare our model against three categories of graph representation learning methods: (1) Homogeneous-**node2vec** [13], **GCN** [60] and **GAT** [57]. (2) Heterogeneous-**mp2vec** [8], **HAN** [58], **HetGNN** [63], **HGT** [17] and **HALO** [1]. (3) Misinformation detection-**FANG** [35] and **MetaHG** [40].

Table 2: Experimental results (%) for the public opinion concern prediction task.

| Dataset | Metrics | node2vec | GCN | GAT | mp2vec | HAN | HetGNN | HGT | HALO | FANG | MetaHG | POFD |
|---------|---------|----------|-----|-----|--------|-----|--------|-----|------|------|--------|------|
| BuzzFeed | AUC | 76.12±0.37 | 92.87±0.58 | 93.29±0.53 | 86.13±0.33 | 93.43±0.51 | 93.77±0.68 | 93.88±0.73 | 94.32±0.48 | 90.92±0.61 | 93.07±0.43 | **96.36±0.33** |
| | AP | 71.46±0.27 | 92.44±0.49 | 92.06±0.39 | 85.93±0.34 | 91.95±0.57 | 92.39±0.58 | 92.49±0.62 | 92.98±0.36 | 90.12±0.41 | 91.68±0.29 | **96.09±0.36** |
| PolitiFact | AUC | 74.22±0.35 | 93.02±0.24 | 93.61±0.64 | 88.16±0.39 | 93.88±0.24 | 93.29±0.35 | 92.39±0.47 | 93.57±0.32 | 91.31±0.29 | 92.85±0.43 | **95.44±0.37** |
| | AP | 69.19±0.37 | 92.49±0.33 | 92.72±0.32 | 89.24±0.43 | 93.21±0.27 | 92.43±0.31 | 92.03±0.23 | 92.75±0.27 | 91.49±0.41 | 92.37±0.38 | **95.57±0.33** |
| Twitter | AUC | 79.14±0.41 | 93.12±0.41 | 93.68±0.45 | 88.30±0.33 | 93.90±0.62 | 92.41±0.43 | 93.12±0.26 | 94.25±0.34 | 92.03±0.39 | 93.12±0.44 | **96.18±0.44** |
| | AP | 78.67±0.24 | 92.50±0.34 | 92.74±0.42 | 87.84±0.45 | 92.82±0.51 | 92.01±0.38 | 92.50±0.34 | 92.41±0.27 | 91.66±0.38 | 92.50±0.33 | **94.99±0.25** |

***Parameter Settings***. We set the embedding dimension to 256, return parameter to 1, in-out parameter to 1, path length to 40, and window size to 10 in node2vec and mp2vec. All GNN models are two-layer structures with a hidden layer dimension of 256 and an output dimension of 128 and are implemented through the PyTorch Geometric (PyG) package [10]. The models that require setting the number of attentional heads are set to 4. In POFD, we select the top 500 user nodes in degree ranking as $U_h$. The energy amplification factor of the fake news event is set to 3. For all neural network models, the learning rate is 0.001, the number of training epochs is 200. For all methods, we run 10 times with the same partition and report the average results. The dataset and code with running instructions are available at `https://github.com/ki-ljl/POFD`.

## 4.2 Public Opinion Concern Prediction

***Setting.*** In this paper, we define public opinion concern prediction as two types of link prediction: **user-user** and **user-event**. Specifically, we randomly sample 10% of the two types of links from the dataset to generate test edges. In the test edges, the positive and negative edges are each 50%. Then, we mask the positive test edges from the dataset to prevent information leakage. We report the AUC and average precision (AP) of all models in Table 2.

***Analysis.*** It is clear that POFD consistently performs better than all baselines on three datasets. Compared to the best performance of baselines, POFD achieves a high improvement in both AUC and AP, which indicates the effectiveness of considering public opinion field effect and social circle influence in representation learning. We also observe that GNNs generally achieve better performance than random walk-based methods, due to the fact that random walk-based methods can only simply exploit structural information in the PON without considering feature information that is crucial in topic diffusion.

## 4.3 Event Classification

***Setting.*** We also conduct the event classification task to evaluate the embeddings learned from POFD. Specifically, since the properties of the event (positive or negative) are unknown, we first set the energy amplification factor of all `poc` nodes to 1. Then for each dataset, we randomly split event nodes into 70%, 10%, and 20% for training, validation and testing. We report the accuracy and F1 scores of all models in Table 3.

***Analysis.*** Based on the results reported in Table 3, we observe that POFD achieves the best performance on all metrics for all three datasets. By capturing the interactions between pofs of different events, POFD can learn more meaningful event node embeddings and thus achieves the best performance on the event classification task.

## 4.4 Universality Analysis

In addition to pof in PON, there are various fields in real life, such as ***academic field*** in DBLP [12]. Specifically, DBLP is a well-known heterogeneous academic network containing four types of nodes (**author**, **paper**, **term** and **conference**) and three types of links (**author-paper**, **paper-term** and **paper-conference**). More information about DBLP can be found in ***Appendix*** A.1. Like events attract users in PON, conference nodes in DBLP attract the attention of authors and their papers (and terms). Specifically, authors may be attracted to multiple conferences (this attraction is linked through author-paper and paper-conference).

***Setting.*** In this experiment, we let the conference nodes be `poc` nodes and then conduct the research domain classification task for the author nodes. We use the DBLP dataset provided in PyG [10], which

Table 3: Experimental results (%) for the event classification task.

| Dataset | Metrics | node2vec | GCN | GAT | mp2vec | HAN | HetGNN | HGT | HALO | FANG | MetaHG | POFD |
|---|---|---|---|---|---|---|---|---|---|---|---|---|
| BuzzFeed | ACC | 77.78±0.38 | 86.11±0.56 | 88.89±0.32 | 80.56±0.37 | 86.11±0.37 | 83.33±0.48 | 86.11±0.51 | 86.11±0.28 | 88.89±0.34 | 88.89±0.41 | **94.44±0.34** |
| | F1 | 69.23±0.51 | 82.76±0.34 | 85.71±0.27 | 72.00±0.46 | 82.76±0.31 | 78.57±0.26 | 82.54±0.43 | 83.33±0.48 | 83.33±0.41 | 84.62±0.35 | **92.31±0.47** |
| PolitiFact | ACC | 70.83±0.31 | 85.42±0.34 | 85.42±0.44 | 72.92±0.47 | 87.25±0.25 | 87.50±0.34 | 89.58±0.38 | 91.67±0.35 | 89.58±0.33 | 93.75±0.25 | **95.83±0.38** |
| | F1 | 66.67±0.37 | 84.44±0.38 | 84.44±0.18 | 72.35±0.44 | 86.06±0.48 | 87.50±0.38 | 88.37±0.34 | 90.48±0.29 | 87.18±0.33 | 92.68±0.39 | **95.24±0.32** |
| Twitter | ACC | 61.14±0.49 | 68.24±0.34 | 68.72±0.27 | 66.82±0.35 | 72.35±0.43 | 72.22±0.43 | 77.78±0.38 | 82.93±0.33 | 78.67±0.33 | 83.33±0.24 | **85.31±0.48** |
| | F1 | 61.32±0.34 | 59.88±0.24 | 62.06±0.54 | 64.29±0.41 | 74.15±0.17 | 66.67±0.27 | 71.43±0.52 | 74.33±0.31 | 72.45±0.43 | 77.34±0.32 | **80.57±0.34** |

Table 4: Experiment results (%) on the DBLP for the node classification task.

| Metrics | node2vec | GCN | GAT | mp2vec | HAN | HetGNN | HGT | HALO | FANG | MetaHG | POFD |
|---|---|---|---|---|---|---|---|---|---|---|---|
| Macro-F1 | 83.67±0.34 | 91.89±0.32 | 90.80±0.42 | 87.42±0.24 | 92.43±0.38 | 92.64±0.45 | 92.01±0.37 | 92.81±0.23 | 90.06±0.30 | 91.13±0.37 | **95.03±0.24** |
| Micro-F1 | 85.27±0.41 | 92.62±0.24 | 91.84±0.37 | 87.92±0.34 | 92.85±0.34 | 92.09±0.37 | 92.60±0.24 | 93.34±0.24 | 90.75±0.25 | 91.74±0.25 | **95.33±0.42** |

(a) Public opinion concern prediction.      (b) Event classification.

Figure 3: Ablation study on BuzzFeed dataset.

contains a predefined training, validation, and testing set. We report the Macro-F1 and Micro-F1 of all models in Table 4.

***Analysis.*** As shown in Table 4, POFD achieves the best performance on DBLP. The experimental results indicate that the ***academic field*** formed by conference nodes can effectively capture the potential influence of conference (poc) nodes on authors and their published papers, which in turn can indirectly guide the classification task of author nodes through conference information (authors always tend to publish in conferences related to their research fields).

## 4.5 Multi-hop Public Opinion Field

In this section, we experimentally demonstrate the reasonableness of our choice of 1-hop neighbors in Definitions 2 and 3. We give experimental results for 1-hop to 4-hop neighbors in ***Appendix*** A.3. The experimental results show that the use of multi-hop neighbors does not improve the model performance, probably because multi-hop neighbors introduce more redundancy. This indicates that pof energy is mainly determined by the 1-hop neighbors of poc nodes.

## 4.6 Centrality

In Eq. (1), we define $w_{i,j} = \lambda_j \cdot \text{att}(\boldsymbol{h}_i, \boldsymbol{h}_j)$, where $\lambda_j$ is used to measure the influence of node $v_j$ in the PON. In this paper, we let $\lambda_j$ be the degree centrality of $v_j$. It is well known that there are various other centrality metrics such as Eigenvector Centrality. However, as we stated earlier, there have been many studies on how to measure the influence of nodes in a network. $\lambda_j$ is only used as an input for the study in this paper, and how to choose $\lambda_j$ is not part of the study in this paper. In this section, we investigate the impact of different centrality metrics on the model performance. The experimental results and analysis can be found in ***Appendix*** A.4.

## 4.7 Ablation Experiment

***Setting.*** In this section, we investigate how the three components of POFD affect its performance. Specifically, we create the following ablations: (1) **POFD**$_{tp}$: keep topic propagation but remove social circle influence. (2) **POFD**$_{sci}$: keep social circle influence but remove topic propagation (we use GAT [57] to replace topic propagation). (3) **POFD**$_{loss}$: remove unsupervised proximity loss. We conduct ablation experiments on BuzzFeed, and the experiment results are shown in Fig. 3.

*Analysis.* Fig. 3 reports the results of the ablation experiments. We observe that POFD outperforms the three variants mentioned above, which indicates that all three modules of POFD can individually improve the model performance.

## 5  Related Work

**Content-Based.** The content-based methods focus on the difussion of information from blogs or news content. Mahdikhani et al. [32] collected over 1.25 million tweets containing timestamps to predict the popularity of tweets (based on the number of retweets) by extracting the features of the tweets. Tsagkias et al. [54] used a two-stage binary classification to predict the number of comments on online news reports. Kaltenbrunner et al. [21] proposed a growth model for predicting the popularity of slashdot stories. Lee et al. [26] investigated the probability that a web content will continue to be followed by online readers after a period of time. In general, this type of approach often requires the processing of content to produce manual features, often with significant workload.

**Homogeneous Network-Based.** With the rise of graph mining techniques [37, 13, 60, 15, 57], a large number of people have started to study how to model information diffusion using graph structures. Bourigault et al. [5] embedded nodes in a continuous latent space and calculate the contamination scores of nodes activated by source nodes through a diffusion kernel function in the latent space. Inf2vec [9] studied the problem of influence embedding of nodes in social networks. RNe2Vec [46] estimated the information diffusion probability based on the random wandering method. Based on the state of the nodes' network and local neighborhoods, DeepInf [41] aimed to predict the activation probability of the nodes. CasSeqGCN [59] combined network structure and time series to predict the information cascade. None of these models can model complex social contexts such as user-event relationships, and thus cannot extract high-quality embedding representations.

**Heterogeneous Network-Based.** The heterogeneous network-based methods learn high-quality embedding representations through diverse neighborhood relationships. Although earlier studies such as [19, 38, 39, 49] also considered heterogeneous scenarios, they all optimized a specific objective (e.g., fake news detection) rather than extracting abundant node representations, and thus were not very generalizable. FANG [35] improved the quality of node representation learning by capturing the rich social interactions between users, news and media to improve the accuracy of fake news detection. Such approaches mainly improve the embedding representation by considering heterogeneous information in the network, but they do not make use of some deeper semantics such as public opinion field effects.

## 6  Conclusion

In this paper, we propose for the first time to fuse the public opinion field effect into the representation learning of trending topics diffusion. Then, we propose a representation learning framework POFD that includes both topic propagation and social circle influence. Experimental results including public opinion concern prediction and event classification demonstrate the effectiveness of POFD. We also show that POFD is also effective for other types of fields (e.g., academic fields in DBLP).

**Ethics Statement.** An important application of our study is fake news controlling. For example, wiping out $130 billion in stock value after a false tweet claimed that Barack Obama was injured in an explosion [42]. Our intention behind this work is that studying trending topic diffusion can help timely discover potential crises caused by fake news and clarify the truth with evidence as soon as possible to avoid unnecessary losses.

**Limitations and Future Work.** We note that POFD does not consider the dynamics of PON, such as the disappearance of old events and the emergence of new events. Therefore, in the future we will investigate representation learning for dynamic PONs.

## Acknowledgement

This work was supported by the State Key Laboratory of Communication Content Cognition Funded Project No. A32003, National Natural Science Foundation of China No. U22A2025 and No. 61972275.

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

# A Appendix

## A.1 Datasets Processing Details

In this section, we describe how the dataset is processed to obtain the inputs for the model.

- **BuzzFeed and PolitiFact**. For the **news** nodes, we use the BERT [7] to process the textual content of the news and generate an embedding of dimension 768 for each news node. For **user** nodes and **source** nodes, we use metapath2vec [8] to generate embeddings of dimension 128 for both types of nodes since there is not enough information as input.

- **Twitter**. We follow the settings in FANG [35] in this paper. For the **news** nodes, we construct a TF.IDF [43] vector from the text body of the article. We enrich the representation of news by weighting the pre-trained embeddings from Glove [36] of each word with its TF.IDF value forming a semantic vector. Finally, we concatenate the TF.IDF and semantic vector to form the news article feature vector. We calculate the **user** vector as the concatenation of a pair consisting of a TF.IDF vector and a semantic vector derived from the user profile's text description. Similarly to article representations, for each **source** node, we construct the source feature vector as the concatenation of its TF.IDF vector and its semantic vector derived from the words in the homepage and the "About Us" section, as some fake news websites openly declare their content to be satirical or sarcastic.

- **DBLP**. In the universality analysis experiment, we directly use the DBLP dataset provided in PyG [10]. The dataset contains 4057 authors, 14328 papers, 7723 terms, and 20 publication venues. The authors are divided into four research areas (Database, Data Mining, Artificial Intelligence, and Information Retrieval). Each **author** node is described by a bag-of-words representation of their paper keywords. For semi-supervised learning models, the author nodes are divided into training, validation, and testing sets of 400 (9.86%), 400 (9.86%), and 3257 (80.28%) nodes, respectively. Since there are no initial features for the **publication venue** nodes in this dataset, we use the torch.randn function to generate initial features of dimension 128 for each publication venue node.

## A.2 Dataset Attributes

We describe some basic properties of the three datasets in Table 1, such as the number of nodes and the number of edges. In addition, we also give the **Average Degree**, **Connectivity** and **Average Clustering Coefficient** of the three networks. Furthermore, we give the degree distribution of the three networks in Fig. 4.

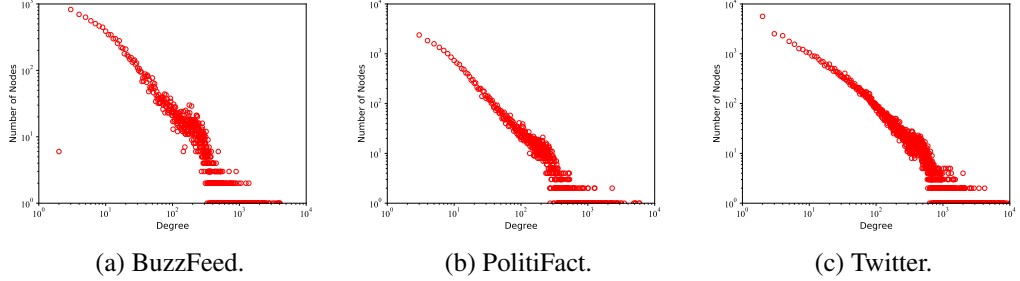

(a) BuzzFeed.        (b) PolitiFact.        (c) Twitter.

Figure 4: The degree distribution of the datasets.

## A.3 Multi-hop Public Opinion Field

In this section, we investigate the impact on model performance when constructing pof and pof energy using multi-hop neighbors of poc node. We give experimental results for 1-hop to 4-hop neighbors in Table 5. Experimental results show that using multi-hop neighbors only degrades the performance of the model. We speculate that using multi-hop neighbors may bring additional redundant information.

## A.4 Degree Centrality and Eigenvector Centrality

We describe the effect of degree centrality and eigenvector centrality on model performance in Table 6, where $POFD_d$ and $POFD_e$ denote the POFD using degree centrality and eigenvector centrality, respectively. Based on the experimental results, we find that the performance of the models implemented with the two centrality metrics is approximately equivalent. We hope that there will be follow-up work in the future to investigate how to assess the importance of nodes in the public opinion space.

Table 5: Multi-hop neighborhood experiment.

| Dataset | Metrics | 1-hop | 2-hop | 3-hop | 4-hop |
|---|---|---|---|---|---|
| BuzzFeed | AUC | **96.36±0.33** | 93.18±0.73 | 89.41±0.33 | 82.72±0.62 |
| | AP | **96.09±0.36** | 93.38±0.66 | 90.32±0.36 | 79.22±0.43 |
| PolitiFact | AUC | **95.44±0.37** | 93.64±0.51 | 88.98±0.37 | 79.83±0.62 |
| | AP | **95.57±0.33** | 94.43±0.23 | 90.55±0.34 | 78.89±0.34 |
| Twitter | AUC | **96.18±0.44** | 91.22±0.26 | 87.34±0.51 | 79.28±0.28 |
| | AP | **94.99±0.25** | 91.53±0.34 | 88.21±0.25 | 80.40±0.33 |

Table 6: Centrality experiment.

| Dataset | Metrics | $POFD_d$ | $POFD_e$ |
|---|---|---|---|
| BuzzFeed | AUC | 96.36±0.33 | **96.89±0.33** |
| | AP | 96.09±0.36 | **96.75±0.28** |
| PolitiFact | AUC | 95.44±0.37 | **95.72±0.41** |
| | AP | **95.57±0.33** | 95.51±0.28 |
| Twitter | AUC | **96.18±0.44** | 95.86±0.51 |
| | AP | **94.99±0.25** | 94.13±0.25 |

