# OpenReview forum: "Public Opinion Field Effect Fusion in Representation Learning for Trending Topics Diffusion"
_NeurIPS.cc/2023/Conference — NeurIPS 2023 poster_

### Official Review · Reviewer_kWLJ · 2023-07-05

**Soundness:** 2 fair
**Presentation:** 2 fair
**Contribution:** 3 good
**Rating:** 5
**Confidence:** 4

**Summary:**

This paper proposes to model the public opinion field effect in representation learning of social networks. The authors propose to model social networks as heterogeneous graphs and incorporate three effects into a neural architecture. Experiments on three datasets demonstrate that the proposed approach outperform several baselines across two classification settings.

**Strengths:**

+ modeling public opinion on social networks is an important research question
+ the connection to media and communications literature is good

**Weaknesses:**

- "studying the diffusion and prediction of trending topics can effectively help governments ... discover potential public opinion crisis and deal with it to avoid their reputation ... losses". This framing in lines 23-24 potentially has great ethical concerns regarding the task and problem.

- In definition 2.3, I understand that only one-hop neighborhood of a center node could contribute to its energy. Why wouldn't 2-hop or more distant entities on social networks contribute to the popularity of certain issues or opinion leaders? They might not directly follow them, but may offer commentary on their own account. That being said, I wonder if the authors have good reasons for why only considering the 1-hop proximity is enough.

- I suggest moving Figure 1 from the appendix to the main paper. It is usually a good idea to present a high-level illustration of the proposed approach.

- My greatest concern is about the selection of baselines. While 9 baselines approaches, mostly GNNs or heterogenous GNNs, are considered, they are somewhat outdated. For example, the most recent baseline is HGT proposed back in 2020. If the authors agree that the field of heterogeneous GNNs or modeling social networks with GNNs have advanced since 2020, it is advised to compare with more recent approaches. Also, the baselines considered in this work are mostly generic GNNs, while more domain-specific methods on misinformation [1], Twitter [2], other social media [3], and more might also be considered or at least discussed.

- I suggest adding statistical significance tests to results in Tables 2-4, especially given that some of the numbers are close.

- I suggest adding an ethics statement / broader impact section since modeling and monitoring public opinions could have ethical concerns and dual use scenarios.

[1] Nguyen, Van-Hoang, et al. "Fang: Leveraging social context for fake news detection using graph representation." Proceedings of the 29th ACM international conference on information & knowledge management. 2020.

[2] Feng, Shangbin, et al. "TwiBot-22: Towards graph-based Twitter bot detection." Advances in Neural Information Processing Systems 35 (2022): 35254-35269.

[3] Qian, Yiyue, et al. "Distilling meta knowledge on heterogeneous graph for illicit drug trafficker detection on social media." Advances in Neural Information Processing Systems 34 (2021): 26911-26923.

**Questions:**

please see above

**Limitations:**

There is a brief discussion in Section 6.

---

> ### Author Rebuttal · Authors · 2023-08-07
>
> We sincerely thank the reviewer for the positive comments on the novelty and importance of our research. We also appreciate the reviewer's suggestion on ethics statement, $k$-hop neighborhood, and more baseline models. We respond carefully to the reviewer’s questions and hope our answers could address your concerns.
>
> **[W1&W6]: Ethical Concerns**
>
> Thanks for your invaluable suggestion. We do understand the reviewer's concern. We have provided a detailed response to this issue in **[Common Q4]** of the global response.
>
> **[W2]: k-hop Neighbors**
>
> Thanks for your insightful suggestions. The main reason of considering 1-hop neighbors is **they are the closest neighbor to the poc node**. For example, people who directly post comments on a hot event or the followers of a famous public figure. These people are the most active ones prefer to participate in the trending topic about their corresponding poc node (e.g., hot event or public figure). There are also indeed someone know a trending topic indirectly or post their comments on a retweeted content. **This phenomenon has been taken account into our social circle influence representation in Section 3.2**.
>
> Moreover, our model can be easily extended to handle multi-hop neighbors by the following equation $E(P_i^{(k)})=\sum_{t=1}^k\sum_{d_{i,j}=t}w_{i,j}\tau_{i, j}$, where $d_{i,j}$ denotes the distance between nodes $v_i$ and $v_j$. We also add a new set of experiments to evaluate the performance by varying $k$ from 1 to 4. As shown in **Table 6** in PDF file, the experimental results show that using multi-hop neighbors does not improve the performance. We are sorry about that we did not explain this point clearly and will give more detailed discussion in our revised paper.
>
> **[W3]: Framework of POFD**
>
> Thanks a lot for your invaluable suggestion. Due to space limitation, we only give the figure to illustrate the public opinion field effect (i.e., Fig.1 in the original paper) and put the figure of framework of the model into supplementary material. We will add the framework diagram of the model in the final version of the paper.
>
> **[W4]: More Baseline Models**
>
> Thanks a lot for your invaluable suggestions.  We add **FANG[1]** and **MetaHG[2]** as baseline models in our experiments. In addition, we also add the latest heterogeneous graph neural network, **HALO[3]**, in our comparison model. We perform a grid search on the parameters of all three models to get the best performance. The experimental results are shown in **Tables 1-3** in the PDF file, which validates that our models outperform the best known models. Since **TwiBot-22[4]** is a public dataset standard rather than a model, we do not add it in our baseline model.
>
> **[W5]: Significance Test**
>
> Thanks a lot for your invaluable suggestion. We conduct T-tests experiments as we mentioned in **[Common Q3]** of the global response. The experimental results validate the superiority of our model. We will add discussion about significance test in revised paper.
>
> **References**
>
> [1] Nguyen, Van-Hoang, et al. "Fang: Leveraging social context for fake news detection using graph representation." Proceedings of the 29th ACM international conference on information & knowledge management. 2020.
>
> [2] Qian, Yiyue, et al. "Distilling meta knowledge on heterogeneous graph for illicit drug trafficker detection on social media." Advances in Neural Information Processing Systems 34 (2021): 26911-26923.
>
> [3] Ahn, Hongjoon, et al. "Descent steps of a relation-aware energy produce heterogeneous graph neural networks." Advances in Neural Information Processing Systems 35 (2022): 38436-38448.
>
> [4] Feng, Shangbin, et al. "TwiBot-22: Towards graph-based Twitter bot detection." Advances in Neural Information Processing Systems 35 (2022): 35254-35269.

---

> > ### Comment · Reviewer_kWLJ · 2023-08-15
> >
> > I would like to thank the authors for the detailed response. I have adjusted my rating accordingly.

---

> > > ### Author Response · Authors · 2023-08-15
> > >
> > > We are very grateful to the reviewer for the recognition of our work and the suggestions are invaluable for improving the quality of our paper. We are very happy to reply your questions about this paper.

---

### Official Review · Reviewer_aU4Y · 2023-07-06

**Soundness:** 3 good
**Presentation:** 2 fair
**Contribution:** 3 good
**Rating:** 5
**Confidence:** 3

**Summary:**

The seminal models of opinion dynamics as information cascades on networks, such as DJ Watts 2002 or Watts and Dodds 2007, uses homogeneous graphs in which the nodes (representing people) are characterized by just one or two numbers (such as a bias number and a Boolean on adoption). This line of research provided a way to quantify the observation that some people are more influential than others, but the simplifying assumptions also limited its success in making predictions. This paper introduces a heterogeneous network setup (with users, events, and sources). It operationalizes three observations about the spread of opinions: an attention competition effect, an effect of popularity on the attention of individual users, an an attentional bias towards negative information. It evaluates the model using user-user link prediction and user-event link prediction on datasets drawn from BuzzFeed, PolitiFact and Twitter, and also with an event-classification task. The authors POFD model achieves very high AUC values for these tasks, improving on a number of older and SOTA baselines.

**Strengths:**

Finding ways to work with heterogenous graphs is an important research area. The definition of the heterogeneous network structure appears well-grounded. It offers an insightful treatment of how opinions promulgated by different opinion leaders can compete for people's attention. Very high performance is achieved on the tasks used in the experiments.

**Weaknesses:**

This is a very technical paper that is written in non-native English, and it is difficult to follow in some places. Having read the paper several times, I conclude that there is a mismatch between the framing of the work and the work itself. The paper succeeds at a static link-prediction task (eg hiding links and then predicting them) and at static event or research topic classification. A dynamic prediction would involve using the state of the network up to a time t_i and using that to predict the state at time t_(i+1) -- possible predictions could be which links are added or which nodes change their state in some other manner. This is a more difficult problem and the paper does not appear to address the growing body of research on the topic.

The paper makes a number of unsubstantiated assumptions, see questions below.

**Questions:**

line 104 onwards. Can you substantiate the assumption that the users breadth of attention is constant?  The total volume of posts of a social media channel fluctuates in response to emergencies and unusual events, indicating that the total attention people pay to public matters may also fluctuate.

line 150 Can you explain why normalized degree centrality is the appropriate measure? For information-propagation problems, eigencentrality may be more relevant. cf SP Borgatti (2005) “Centrality and network flow”, Borgatti and Everett (2006) “A graph-theoretic perspective on centrality”.

line 215 The assumptions about the similarity of users sharing the same opinion seem excessively strong. In other domains, such as marketing, socio-demographic characteristics are only moderate predictors of opinions. Please offer further justification, or admit that this assumption is a limitation of the study.

Section 4.1 Experimental Setup

Are these datasets balanced or not if not, how unbalanced are they? What are the properties of the graphs they define (eg connectivity, distribution of node degrees…)

lihe 277 What’s the justification for these parameter settings? Were the parameters systematically hypertuned? If so, how?

line 311 In what way does research domain classification of authors shed light on the propagation of opinions or topics in research?


**Limitations:**

The authors clearly state in their last sentence that the study does  not consider the dynamics of the PON (the public opinion network). The author(s) indicate an intention to pursue this issue in future research. This is the primary limitation of the study, which should have been represented as study on the coherence or cohesiveness of public opinions (considered as a static problem). The study does not actually address trending, diffusion, or virality of opinions.

The study's general research area is very well established, and presents relatively few societal risks.  Algorithms relating to fake news detection do have the potential for societal harm, if deployed by platform moderators or in government surveillance, because both false negatives and false positives can be very harmful (by allowing fake news to spread, or suppressing legitimate free speech). The authors do not discuss this potential harm. The work does not risk this harm any more than the large body of related work, however.

---

> ### Author Rebuttal · Authors · 2023-08-07
>
> We sincerely thank the reviewer for positive comments on the novelty and high performance  of our work. We respond carefully to the questions and hope our answers could address your concerns.
>
> **[W1&L1]: Dynamic of trending topic diffusion**
>
> Thanks a lot for your insightful suggestion for considering dynamic of trending topic. As mentioned by reviewer, the information diffusion prediction is always based on information cascade, which is to identify potential users who are likely to participate in information sharing at next time. It is a dynamic process. However, the main objective of this work is not to study the cascade representation but to investigate users’ representation by considering public opinion field effect. We study the influence of users’ attention distribution in our representation model when several trending topics coexist in public opinion space at the same time. **In practice, the trending topics do not alternate frequently in a short period of time and thus the existence of trending topics is relatively stable. Our users’ representation can be utilized as part of cascade-based prediction model for dynamic information diffusion**. Information diffusion prediction is a downstream task and our representation model can help existing methods to significantly improve the performance of prediction.
>
> The “dynamics of the PON” we mentioned in last sentence means **the popularity change of trending topics in public opinion space**. Every trending topic has the process of breaking out, developing and extinction. The popularity of trending topic always changes and interacts with users’ participation. We will study the effect of popularity dynamic in our future work. It is a very difficult but interesting problem and we hope this paper could inspire more researchers pay attention to this problem.
>
> **[Q1]: Attention is Constant**
>
> We sincerely thank you for reading our paper carefully and it is a very good question. Several studies [1-3] on social media substantiate the attention amount is limited and thus we introduce “Attention competition effect” in our representation learning model to reflect that how is users’ attention distributed when several hot topics coexist in social networks. We agree with that when an emergency or unusual event happens, it will attract a large amount of attention from people. On one hand, if there is no other hot topic in public opinion space, our model can handle this case by considering that there is only one POF of this emergency or unusual event. On the other hand, if there have existed some other hot topics before this emergency or unusual event happening, the attention will be distracted from previous hot topics. Our model also can handle this case by considering several POFs. One of the main contributions of this work is to investigate how to obtain a high-quality representation when there are several coexisting hot topics in public opinion space. We are sorry about this point is not clarified clearly and we will give more discussion in revised paper.
>
> [1] Limited individual attention and online virality of low-quality information. Nature Human Behaviour, 2017
>
> [2] The spread of true and false news online. Science, 2018
>
> [3] Accelerating dynamics of collective attention. Nature communications, 2019
>
> **[Q2]: Degree Centrality and Eigenvector Centrality**
>
> We sincerely thank you for your insightful suggestion. Actually, influence measure $\lambda_i$ can be considered as an input of our model. As we mentioned in line 90 and 150, existing studies have proposed various influence measures. For ease of understanding $\lambda_i$, we explain it and use normalized degree centrality as influence measure in this paper, because intuitively a blogger has more influence if he/she has more fans. We also agree with the reviewer's suggestion that eigenvector centrality is better and we add new experiments in **Table 5** to investigate performance under different kinds of influence measure. We will explain it more clearly in revision.
>
> **[Q3]: Reasonability of the Similarity Assumption**
>
> Thanks for your question. The similarity we mentioned here means that **people who surrounding the same poc node has the similar preference to participate in the discussion or propagation of the corresponding trending topic but not they have the same opinion**. For example, the fans of an idol are willing to discuss the hot topic about this idol. **We reinforce similarity only to better predict users' willingness to engage in something, not to reinforce the overall similarity among users**. We add new ablation experiments about this similarity. From **Figure 1** in the PDF, we find that the similarity assumption within POF can indeed improve the accuracy of the model. We are sorry about the misunderstanding caused by this “similarity”, we will give more details in revised paper.
>
> **[Q4]: Attributes of Datasets**
>
> Thank you for your suggestions. **Table 4 and Figure 2** in PDF file depict the **average degree, connectivity, average clustering coefficients, and degree distributions** for the datasets. We will give more details about the three datasets in the revision.
>
> **[Q5]: Experimental Parameter Settings**
>
> Thank you for your suggestions. We give details of the parameter settings in **[Common Q2]** of the global response and we will add these details in revised paper.
>
> **[Q6]: DBLP**
>
> Thanks for your suggestion. We would like to demonstrate through DBLP experiments that **the field effect does not just exist in social networks, but is ubiquitous**. For DBLP dataset, conference nodes may form an academic field to compete researchers' attention, and author nodes outside the field will connect themselves to the conference topic through paper nodes inside the field, which can be used to infer the authors' research interest. Therefore, it is not for topic propagation prediction but research interest inference on DBLP dataset. We will explain it more clearly in the revision.

---

> > ### Comment · Reviewer_aU4Y · 2023-08-10
> > **Thanks for answers**
> >
> > Thank you for your answers to my questions. In the event that the paper is accepted, I strongly recommend that you clarify the motivation and framing of the work so that the reader does not expect results about the dynamics (over time).

---

> > > ### Author Response · Authors · 2023-08-11
> > >
> > > Thanks for your reply. We are very happy to be able to communicate with you about this work, which really improves the strength of our paper, whether it is accepted or not. If the paper is accepted, we will explain this part in detail in the final version.

---

### Official Review · Reviewer_1ymS · 2023-07-23

**Soundness:** 3 good
**Presentation:** 3 good
**Contribution:** 3 good
**Rating:** 6
**Confidence:** 4

**Summary:**

In this paper, the public opinion field effect is fused into a representation learning framework for trending topics diffusion. The proposed algorithm includes representation learning for both topic propagation and social circle influence. Experimental results on public opinion concern prediction and event classification demonstrate the effectiveness of the proposed algorithm.




**Strengths:**

1. The universality analysis and the ablation study in this paper provide insights such as the proposed algorithm is applicable to academic field in DBLP, and topic propagation plays a major role in the event diffusion.

2. The complexity analysis shows the proposed representation learning algorithm is efficient.

3. This is the first work to consider these public opinion field  effects in representation learning for trending topic diffusion, and the ideas to incorporate “Attention Competition Effect”, “Popularity Dominance Effect” and “Negative Bias Effect” in representation learning are novel.



**Weaknesses:**

1. Are the comparison results in Table 2, 3, 4 statistically significant? T-tests can be conducted.



**Questions:**

NA

---

> ### Author Rebuttal · Authors · 2023-08-07
>
> We sincerely thank the reviewer for the recognition of the novelty and importance of our work. We also appreciate the positive comment on the effectiveness and efficiency of our model. We respond carefully to the reviewer’s questions and hope our answers could address your concerns.
>
> **[W1]: Significance Test**
>
> Many thanks to the reviewer for the suggestion. As stated by the reviewer, in order to make our experimental results more credible, we perform T-tests on multiple sets of experimental results. Specifically, we use the **test_rel()** function in the scipy package to calculate the p-value of the result array of our model and the result array of the baseline model, and determine whether the difference between our model and the comparison model is statistically significant by whether the p-value is less than 0.05.
>
> We present the experimental results in **Tables 1-3 in the PDF** file of the global responses. **"+"** in the table indicates that the p-value is less than 0.05, i.e., it is acceptable that our model significantly outperforms the baseline models. **The experimental results show that our model significantly outperforms all the baseline models**.

---

> > ### Comment · Reviewer_1ymS · 2023-08-12
> > **thank the authors for the response**
> >
> > Thanks a lot for the follow up study on t-tests. The results make sense.

---

> > > ### Author Response · Authors · 2023-08-12
> > >
> > > Thanks for your reply. Your suggestions have greatly improved the quality of our paper.

---

### Official Review · Reviewer_7Tx8 · 2023-07-24

**Soundness:** 3 good
**Presentation:** 4 excellent
**Contribution:** 2 fair
**Rating:** 5
**Confidence:** 3

**Summary:**

In the manuscript, the authors consider two interesting novel factors for trending topics diffusion, which are public opinion field effect and social circle influence effect. They propose a representation learning framework to incorporate the two effects and conduct extensive experiments to validate the effectiveness of the proposed model.



**Strengths:**



The motivation is clear and the manuscript is easy to follow. The consideration of the two effects in representation learning for trending topics diffusion is novel. The technique part seems to be reasonable.

**Weaknesses:**

The authors introduce three specific kinds of public opinion effect, “Attention Competition Effect”, “Popularity Dominance Effect” and “Negative Bias Effect”, and design the model to obtain the higher quality representation by considering these effects in trending topic diffusion obvia fusing these effects in trending topic diffusion. As there could be many other effects in reality, is the proposed model fixed or can it be extended to other effects? If fixed, the potential of proposed model seems to be limited.


**Questions:**

 The authors introduce three specific kinds of public opinion effect, “Attention Competition Effect”, “Popularity Dominance Effect” and “Negative Bias Effect”, and design the model to obtain the higher quality representation by considering these effects in trending topic diffusion obvia fusing these effects in trending topic diffusion. As there could be many other effects in reality, is the proposed model fixed or can it be extended to other effects?


**Limitations:**

The authors introduce three specific kinds of public opinion effect, “Attention Competition Effect”, “Popularity Dominance Effect” and “Negative Bias Effect”, and design the model to obtain the higher quality representation by considering these effects in trending topic diffusion obvia fusing these effects in trending topic diffusion. As there could be many other effects in reality, is the proposed model fixed or can it be extended to other effects? If fixed, the potential of proposed model seems to be limited.

---

> ### Author Rebuttal · Authors · 2023-08-07
>
> We sincerely thank the reviewer for positive comments on novelty, good presentation and reasonable technique of our work. We also appreciate the reviewer's suggestion about giving more discussion on  public opinion field effect. We respond carefully to the reviewer’s questions and hope our answers could address your concerns.
>
> **[W1&Q1&L1]: More Discussion about Public Opinion Field Effect**
>
> Many thanks to the reviewer for your insightful questions. In this paper, we focus on the allocation of people's attention when multiple trending topics coexist. Therefore, we choose the three effects that are most relevant to attention allocation without considering the other effects. For a more detailed explanation, please refer to **[Common Q1]** in the global response.
>
> **However, we would like to clarify that our model is not fixed, rather it can be easily extended to some common known effects**. As we stated in **[Common Q1]**, the $\textbf{\textit{Broken Window Effect}}$ is similar to the $\textbf{\textit{Negative Bias Effect}}$ and the $\textbf{\textit{Snowball Effect}}$ is similar to the $\textbf{\textit{Popularity Dominance Effect}}$. Therefore, our model can be easily extended to these effects that are similar in meaning.
>
> In addition, our model can be easily extended to some effects that do not apply to attention allocation. An example is the $\textbf{\textit{Primacy Effect}}$ (People have inherent stereotypes about certain groups of people). If we want to investigate the effect of stereotypes on information diffusion, we can make the following changes based on the model in this paper:
> 1. Certain nodes in the public opinion network are classified into groups using known information or graph clustering techniques.
> 2. Representative features of each group are obtained using topic extraction or other relevant techniques. For example, the spread of fake news in social networks makes people no longer trust the news media, and many people label the news media as "fake". Through feature extraction, we can get the overall "fake" features of the news media group.
> 3. In the representation learning stage, in addition to aggregating the features of the public opinion field where the neighbor nodes are located, **the user will also aggregate the features of the group where the neighbor nodes are located, in order to simulate the user's inherent impression of the neighbor**, i.e., the $\textbf{\textit{Primacy Effect}}$.
>
> In summary, since we only study the problem of attention allocation, we only consider the three effects mentioned in the paper. However, our model has a good scalability. In the future, we will consider incorporating other effects into our model.

---

### Official Review · Reviewer_NVSH · 2023-07-25

**Soundness:** 2 fair
**Presentation:** 3 good
**Contribution:** 3 good
**Rating:** 5
**Confidence:** 2

**Summary:**

This paper proposes a Representation Learning method that considers field information in topic diffusion, i.e., situations with multiple topics and opinion leaders. It defines new concepts (public opinion network/field and public opinion field energy) and proposes a method for incorporating some observations into Representation Learning. It has been shown to perform better than conventional methods with multiple tasks and multiple data sets.

**Strengths:**

-  The paper is well-structured and easy to follow.
- The definitions in section 2 are clear, and Figure 1 is helpful in understanding them.
- The validation is performed using multiple real data sets; Although the sources are not specified for the DBLP data sets, they are probably all open data, and the proposed method's source code is publicly available. Hence, they are likely to be reproducible.
- The proposed method outperforms relatively new existing methods in the Public Opinion Concern Prediction task, the Event Classification task, and the research domain classification task.

**Weaknesses:**

- I could not clearly understand how the roles of w={i,j} and tau={i,j} are different. I understand that each is expressed in equation (1), but it isn't easy to understand intuitively. What would tau={i,j} represent in Michael Jordan's example?
- I felt that Observations 1-3 were shown abruptly. Why should these three effects be prioritized and only these be addressed? Are there any other candidate effects that should be treated?
- In the section on experiments, the parameters of the proposed and baseline methods are clearly described, but there is no mention of how these values were determined. I wonder if this is a fair comparison. How were the best parameters determined? Did you search for the best parameters for each method?
- Why are Node2vec and mp2vec less accurate? Other than Node2vec and mp2vec, some methods score close to the proposed method (POFD), but is it possible that the parameter settings and randomness may reverse the superiority of the methods? Statistical tests would be helpful.

Others
- Typo: "Becasue" at P4

**Questions:**

It would be helpful if the authors could answer the points I mentioned in "Weakness".

**Limitations:**

- The authors appropriately mention Limitations.
- As the authors say, they could be used by governments and businesses to anticipate and take action in advance to prevent the spread of information that is inconvenient to them, but they need to be sure that preventing the spread is the ethically correct thing to do.

---

> ### Author Rebuttal · Authors · 2023-08-07
>
> We sincerely thank the reviewer for the positive comments on the strength of experiments and good presentation. We also appreciate the  suggestion about more discussion on public opinion field effect and the fairness of the experiment. We respond carefully to the reviewer’s questions and hope our answers could address your concerns.
>
> **[W1&Q1]: $w_{i,j}$ and $\tau_{i,j}$**
>
> Intuitively, in our model, $\tau_{i,j}$ indicates the preference degree of a user $v_j$ to participate in a topic $v_i$. For example, the users who are interested in basketball are more willing to participate in discussion about the basketball-related topics than those who are not interest in basketball. $w_{i, j}$ reflects how much attention from other people will be brought if a user $v_j$ participate in a topic $v_i$. For example, Michael Jordan is a world-famous basketball superstar. When Jordan talks about some topic on social media, it will bring more public attention (e.g., from his fans ) to this topic than ordinary people. We will explain this point more clearly in revised paper.
>
> **[W2&Q2]: Why prioritise the three effects in the paper? Are there still candidate effects?**
>
> We are very grateful to the reviewers for their questions. We acknowledge that we do not have a corresponding explanation for why we chose the three effects and promise that we will add a reasonable explanation in the final version. A detailed explanation of the reasons for choosing the three effects and whether there are more candidate effects is given in **[Common Q1]** of the global response.
>
> **[W3&Q3]: Fairness of Experiments**
>
> Thanks a lot for your question. We give details of the parameter settings in **[Common Q2]** of the global response. Further, we give the results of the significance tests in **[Common Q3]**. These explanations ensure that our comparisons are fair.
>
> **[W4&Q4]: Node2vec, Mp2vec and Statistical Tests**
>
> In contrast to graph neural network models that utilize both node feature information and network topology, **node2vec and mp2vec can only perform node embedding based on the topology information of the network**, ignoring the important initial node feature information, and thus exhibit poor performance. In order to prove the superiority of our model, we give the significance test in **[Common Q3]** of the global response and the experimental results prove that our model is significantly better than the comparison model.
>
> **[L1]: Ethical Concerns**
>
> Thanks for your suggestion. We do understand the reviewer's concern. We have provided a detailed response to this issue in **[Common Q4]** of the global response.

---

> > ### Comment · Reviewer_NVSH · 2023-08-15
> >
> > Thanks for the response. I believe the author's rebuttal. I have raised my score, believing that the parameter settings in the experiment were fair.

---

> > > ### Author Response · Authors · 2023-08-15
> > >
> > > We deeply appreciate the reviewer for the recognition of our work. It is our pleasure to answer the questions in rebuttal process. Your suggestions are invaluable for us improving our paper.

---

### Author Rebuttal · Authors · 2023-08-07

We would like to thank all the reviewers for their insightful and invaluable comments, which really help to improve the quality of our paper. We also appreciate five reviewers' positive comments about the novelty and well-organization of our paper.  Here, we response to four common questions and hope our answers could address reviewers' concerns. **All the tables and figures involved in rebuttal are summarized in PDF file**, which can be downloaded at the bottom of the global response.

**[Common Q1]: More Discussion about Public Opinion Field Effect**

We investigate how to incorporate public opinion field effect in representation learning for trending topic diffusion. The trending topic diffusion problem essentially is an information diffusion prediction problem, which is an important and fundamental task in social networks and has been extensively studied. The existing works regard information diffusion is an independent process. However, in real world, there are always several trending topics in the social network at the same time. These topics compete for people’s attention and thus information diffusion process will be affected. The existing works neglect the influence of coexisting trending topics. To solve this problem, we introduce three well-known and important public opinion field effects, “Attention Competition Effect”, “Popularity Dominance Effect” and “Negative Bias Effect”, which are most closely related to the allocation of people’s attention in media and communication studies, to investigate how is people’s attention affected by these coexisting trending topics in public opinion field and thus trending topic diffusion can be predicted more accurately.

As the reviewers are concerned, there are some other effects in the real world. However these effects are not closely relevant to the influence of coexisting trending topics on information diffusion. For example, two following effects reflect what are the people’s opinion and emotion:

$\textbf{\textit{Empathy Effect:}}$ People are prone to identify and empathize with events or people similar to themselves.

$\textbf{\textit{Primacy Effect:}}$ People habitually put labels on certain events or groups.

In addition, there also are some other effects, e.g., “$\textbf{\textit{Broken Window Effect}}$”, “$\textbf{\textit{Snowball Effect}}$” and “$\textbf{\textit{Herd Effect}}$” in media and communication studies, but the meaning behind these effects are very similar to three effects introduced in this paper and thus we do not discuss them.

In summary, we study three most closely relevant effects in our representation learning model. There indeed exist various effects in public opinion space, and we cannot discuss all of them in one paper due to our knowledge limitation. **We hope this work can inspire more researcher pay attention to studying public opinion field effect in information diffusion prediction task.**

**[Common Q2]: Experimental Parameter Settings**

Both reviewers NVSH and aU4Y have questions about the details of the parameter settings and the fairness of the experiment. Firstly we apologise for not providing a more detailed description of the parameter settings in the original paper. We have expressed the steps of parameter setting in the experiment as follows, and promise to add this section in the final version.

For all models appearing in the experiments, **we perform a hyperparameter search and select the parameters that perform best on the validation set as the final parameters to ensure that the experiments are fair and comparable**. In particular, we report the best results for all metapaths in mp2vec. For all neural network models, in order to ensure that the experiments are comparable, we set the basic parameters such as number of layers and hiding size to be the same for all models, and then perform a grid search on the other parameters of the models to ensure that we get the optimal performance.

**[Common Q3]: Significance Test**

The reviewers NVSH, 1ymS and kWLJ suggest us to perform some statistical analyses of the experimental results in order to better demonstrate the superiority of our model. We are very grateful to the three reviewers for their suggestions. We conduct t-tests on multiple sets of experimental results, and the results are displayed in **Tables 1-3** in the PDF file. The test results demonstrate that our model significantly outperforms all of the comparative models.

**[Common Q4]: Ethical Concerns**

We greatly appreciate the reviewer’s reminders about ethical concerns that this study could be used to prevent the spread of information which is inconvenient to some organizations. Actually, our study is used for trending topic diffusion. Trending topic diffusion essentially is the problem of information diffusion prediction, which is a fundamental task in social networks and has been extensively studied in recent years. Different from previous studies, we consider public opinion field effects in representation learning model for the first time to make it more applicable to real world. An important application of information diffusion prediction is fake news controlling. For example, wiping out $130 billion in stock value after a false tweet claimed that Barack Obama was injured in an explosion [1]. We are sorry about that the last sentence of first paragraph in introduction leads to the misunderstanding on our study. Our intention behind this sentence is that studying trending topic diffusion can help timely discover potential crises caused by fake news and clarify the truth with evidence as soon as possible to avoid unnecessary losses. We will revise this statement in the revision.

[1] K. Rapoza, “Can ‘fake news’ impact the stock market?” Forbes, 26 February 2017

**The following attached pdf file includes the tables and figures involved in rebuttal.**

---

### Decision · Program_Chairs · 2023-09-21

**Decision:**

Accept (poster)

**Comment:**

This paper proposed a heterogeneous representation learning framework to incorporate public opinion field effect and social circle influence effect. The consideration of the two effects in representation learning for trending topics diffusion is interesting. Experiments were performed on multiple datasets and the proposed method outperforms existing methods. The reviewers raised a few concerns, e.g., unclear technical details or experimental details, lack of significance test, and ethical concerns. The authors made good clarifications in the rebuttal and most reviewers slightly increased the scores. I think most of the issues raised in the reviews are relatively minor issues regarding paper writing, which can be well addressed in the revised version.